# Four Problems in Sexting Research and Their Solutions

**Erin Leigh Courtice**  **and Krystelle Shaughnessy ***

School of Psychology, University of Ottawa, Ottawa, ON K1N 6N5, Canada; ecour048@uottawa.ca
* Correspondence: krystelle.shaughnessy@uottawa.ca

**Abstract:** Despite over 10 years of research, we still know very little about people's sexting behaviours and experiences. Our limited and, at times, conflicting knowledge about sexting is due to re-searchers' use of inconsistent conceptual definitions of sexting, dubious measurement practices, and atheoretical research designs. In this article, we provide an overview of the history of sex-ting research and describe how researchers have contributed to the 'moral panic' narrative that continues to surround popular media discourse about sexting. We identify four key problems that still plague sexting research today: (1) imprudent focus on the medium, (2) inconsistent conceptual definitions, (3) poor measurement practices, and (4) a lack of theoretical frameworks. We describe and expand on solutions to address each of these problems. In particular, we focus on the need to shift empirical attention away from sexting and towards the behavioural domain of technology-mediated sexual interaction. We believe that the implementation of these solu-tions will lead to valid and sustainable knowledge development on technology-mediated sexual interactions, including sexting.

**Keywords:** sexting; cybersex; technology-mediated sexual interaction; computer-mediated communication; research methods; measurement

## 1. Introduction

Over the past decade, many researchers have turned their attention to people's use of communication technology for the purpose of sexual interactions. Both popular media and scholarly discourses have focused on sexting, originally defined as the use of mobile phones to create and exchange sexually explicit content (e.g., via Short Messaging Service or Multimedia Messaging Service [1]). Despite over ten years of empirical studies examining sexting among adolescents and adults, little is known about the predictors, correlates, outcomes, or even the prevalence of sexting. This limited, at times conflicting, knowledge is partly due to researchers using inconsistent measurement practices and definitions of sexting. These inconsistencies have made it difficult to synthesize information about sex-ting across studies, across time, and across technologies [2–8]. Specifically, researchers have and continue to examine sexting in a way that focuses on particular formats, devices, and interaction dynamics, to the detriment of understanding the underlying behaviour that persists as technologies change. These conceptual and methodological limitations have created silos in our broader understanding of the essential behaviour that underlies sexting: sexual interactions via technology. It also limits knowledge about behaviourally similar activities that may actually coincide with or be prompted by "sexting" (e.g., phone sex, cybersex). Addressing these limitations is pressing; technology has already become an integral part of people's communication in an evolving digital world. Thus, researchers must adopt new and more appropriate approaches to understanding sexting and other forms of technology-mediated sexual interaction. In this paper, we review four key problems with the research on sexting: (1) imprudent focus on the medium, (2) inconsistent conceptual definitions, (3) poor measurement practices, and (4) a lack of theoretical frameworks. We describe and expand on solutions to address each of these problems. However, these solutions are unlikely to occur in sexting research; therefore, we recommend that researchers stop studying sexting altogether. Instead, we propose that researchers adopt

the behavioural domain of technology-mediated sexual interaction to replace and improve upon sexting research [4]. We believe that the implementation of these solutions will lead to valid and sustainable knowledge development on technology-mediated sexual interactions, including sexting.

## 2. How Did We Get Here?—A Brief History of Sexting Research

People have long used analog technologies as a medium for engaging in sexual behaviour. For example, the invention of the printing press made the written word more accessible. Specifically, widespread access to the printed word led more people to learn to read and write [9]. Subsequently, more people exchanged erotic letters with romantic and/or sexual partners. Similarly, the invention of photography and film development—including self-developing photo technology—allowed people to take intimate photographs of themselves to share with others. The development and widespread adoption of the telephone provided another new opportunity for sexual communication via technology, with the first 'phone sex' business beginning operation in the 1970s [10]. Indeed, as telephone lines became private and common household technologies, it is likely that people increasingly used them to communicate sexually with others (outside the phone sex business context). However, the exchange of sexual letters, photographs, and sexual conversation via telephone were, and largely remain, relatively taboo activities. Moreover, there remain few empirical studies about these behaviours.

In the mid-1990s, two technological developments ushered in the so-called digital era: the affordability and accessibility of (i) personal computers and (ii) the World Wide Web. These new developments subsequently presented new technologies to facilitate sexual interactions. Alongside the upsurge in computer and internet access, online chat rooms—capable of hosting many different users—became popular, as did people's use of chat forums to specifically engage in computer-mediated sexual interactions with (usually) strangers. As online chat continued to grow in popularity, AOL, Yahoo, and MSN all launched their own respective internet chat environments. People quickly adopted these for interpersonal and sexual interactions with strangers, as well as with in-person acquaintances, friends, and romantic partners. This new medium for sexual interactions was initially described by the popular media as 'cybersex'—a portmanteau of the words 'cybernetics' and 'sex'—referring to sexual activity between two or more people who are connected over a computer network (e.g., the internet). In the late 1990s, researchers began to empirically study cybersex (as well as other online sexual activities), largely describing these activities as 'addictive', 'compulsive', and generally 'problematic' [11–16].

Cybersex is arguably a predecessor of sexting, even though the initial technologies looked quite different. As a term, 'sexting' emerged as a combination of the words 'sex' and 'text messaging' and initially described the exchange of sexually explicit SMS messages between two (or more) people (The origins of this term can be traced to a 2004 article written by Josey Vogels for Canadian newspaper *The Globe and Mail*, about sexually explicit text messages exchanged between David Beckham and one of his assistants. The first peer-reviewed definition of sexting was proposed by Weisskirch and Delevi (2011) [1]). Cell phones with SMS (Short Messaging Service) capability grew in accessibility—and therefore popularity—throughout the early 2000s as 3G (and then 4G) networks expanded throughout Western countries. Indeed, the earliest definitions of sexting focused on the exchange of sexually explicit text messages [17]. However, as mobile technology evolved, so did definitions of sexting. In the later 2000s and early 2010s, cell phones with cameras and MMS (Multimedia Messaging Service) capabilities became accessible and popular. These allowed people to take and send their own photos to others via mobile phone. Alongside these capabilities, the definition of sexting evolved to include sexually explicit photos and videos [18]. The first smartphones appeared on the mass consumer market in 2006 (the first iPhone in 2007), and quickly became popular worldwide as they became more affordable. Smartphones allowed people a mobile connection to the internet, including to websites, messaging services, email, and social media. This device created the opportunity

for people to send messages via traditionally computer-mediated platforms (e.g., email, Facebook Chat), as well as established novel, internet-mediated smartphone applications (e.g., BlackBerry Messenger, WhatsApp). Yet again, the definition of sexting evolved to capture these new mobile capabilities: sending sexually explicit content via any platform that could be accessed via mobile phone [19].

**Sexting as a moral panic**. The media and researcher narratives on sexting follow the trajectory of a typical moral panic [20]. Moral panics are a common response to new technologies [21]. A moral panic is a widespread movement based on an exaggerated perception that a 'deviant' cultural behaviour or group of people threatens the values, interests, or well-being of a community or society [22]. Typically, moral panics are perpetuated by the mass media, and exacerbated by politicians and lawmakers. Indeed, 'sexting' came into widespread cultural prominence alongside a few key events. In 2007, 18-year-old actress Vanessa Hudgens' privacy was compromised when nude photos that she had taken of herself surfaced online and circulated across the internet. After facing criticism from fans and the popular media, Hudgens apologized for taking the photos (despite having had no control over their release). Soon after, in 2008, 18-year-old Jessica Logan died by suicide after her ex-boyfriend shared her self-taken nude images with classmates at their high school. Logan experienced severe bullying from her peers, which people close to her said contributed to her death. The popular media used both of these events to frame sexting as a death sentence, mostly for teenage girls—in Hudgen's case, career death, in Logan's case, death by suicide. Thus ensued the moral panic narrative surrounding sexting, especially among adolescents.

Formal studies on the prevalence of sexting among young people were conducted shortly after these key events. Most of the first studies were conducted by non-academic American organizations [18,23,24]. The first of these studies—conducted in 2008 by CosmoGirl.com in collaboration with The National Campaign to Prevent Teen and Unplanned Pregnancy [18]—found that 20% of teenagers reported they had, at least once, sent or posted a nude or semi-nude picture or video of themselves. The results of this study were widely publicized by mainstream outlets including *The New York Post* ("Sex 'Cells' for Naked Teenagers"), *Chicago Tribune* ("OMG—Sexting is Widespread"), *The Seattle Post* ("Is Sending Racy 'Sexts' Flirting, or is it Porn?"), NPR ('Sexting': A Disturbing New Teen Trend?"), NBC ("Sexting: An Alarming Trend"), and *The New York Times* ("Sexting May Place Teens at Legal Risk"). In late 2009, Google searches for 'sexting' began to steadily climb, culminating in its label as one of *Time Magazine's* 'Top Ten Buzzwords' of that year and a finalist for the *New Oxford American Dictionary's* 2009 Word of the Year (ultimately losing to 'unfriend'). The first academic empirical study was published by Weisskirch and Delevi, three years following the mainstream media attention in 2011 [1]. Many researchers followed suit shortly after.

Researchers did, and largely continue to, examine sexting from the perspective of crisis in a technological, legal, sexual, and moral sense. For instance, researchers often frame teenage sexting as a form of child pornography or as part of a cyberbullying epidemic (Sexual images of underage people can indeed fall within some legal definitions of child pornography; similarly, the non-consensual circulation of sexual images has led to bullying for some adolescents. However, the act of sexting among teenagers does not always constitute 'child pornography' nor does it automatically lead to bullying. Much like other forms of underage sexual activity, these are complex issues and require nuance in all cases). Adult sexting is often discussed in terms of deviance (e.g., infidelity, addiction). Similarly, much of the scholarly literature on sexting assumes that there is something aberrant about the activity. For example, Weiss and Samenow (2010) [25] called for academic research on "Smart Phones, Social Networking, Sexting and Problematic Behaviours"; Wiederhold (2011) [26] asked academics, "Should Adult Sexting Be Considered for the DSM?" Many early sexting researchers examined the prevalence of sexting within particular populations (e.g., adolescents) and focused on the relationships between sexting and other perceived 'risky' or 'unsafe' behaviours (e.g., alcohol consumption, unprotected sex), or undesirable

personality traits (e.g., impulsivity). In fact, Döring (2014) [27] found that 79% of the studies published on sexting emphasized the risks and negative outcomes of sexting. However, more recent empirical studies have suggested that, for many people, sexting can be a form of interpersonal intimacy and communication [4]. Research on the potential benefits of sexting remains limited, particularly relative to the amount of research on the 'problematic' aspects of sexting.

Around the world, research ethics have been guided by reports and declarations such as the Belmont Report [28] and the Declaration of Helsinki [29]. At the core of these and similar guidelines for ethical research are three fundamental principles: respect for persons, beneficence, and justice. To fulfill these baseline ethical standards, researchers must be committed to 'do no harm' and weigh the pros and cons of research projects before conducting them. During moral panics related to new technologies, public concerns are magnified, and academic impact is heightened [30]. Therefore, scientific research that is informed by and predicated upon a moral panic can and does have far-reaching ethical implications. Unfortunately, some sexting research has been used to lend credibility to the 'risk-forward' narrative of sexting, especially about young people sexting. The ethical implications of this are clear; sexting research has had a profound impact on legislation that predominantly serves to threaten people who have sexted with legal penalties (e.g., broad application of child pornography charges in inappropriate contexts [31]). Undoubtedly, the intentions of these laws and related policies are to protect young people from harm. However, they—and the research supporting a one-sided view of sexting—also inadvertently create harm. For instance, Setty (2019) [32] suggested that the indirect impacts of these researcher-informed policies included social shaming and victim blaming of young people who have sexted, as well as a denial of rights to bodily and sexual expression. In the context of broad social pressures such as those created by moral panics, researchers must take additional precautions to consider the potential far-reaching harms to stakeholders prior to undertaking research (including exploratory research). It is now the responsibility of sexting researchers to not conduct and present research from a more balanced perspective, but also promote a balanced perspective amongst policy makers, legislators, educators, and the general public.

The initial moral panic around sexting introduced researchers from multiple disciplines (e.g., psychology, communications, law) to a new area of study that required attention and information, especially in order to offset any potential harms. Because of the perceived urgency around the 'crisis' of sexting, this early research was exploratory and disproportionately focused on establishing the prevalence, 'risks', and negative outcomes of sexting. In combination, the exploratory and risk-forward research indirectly created methodological issues that are still present in current sexting research. Namely, questions developed to measure sexting were created without following proper measure development protocols and were geared towards pre-existing negative biases about sexting. Indeed, the history of sexting and sexting research shed light on the roots of the problems that still plague current sexting research. It is time for researchers to face and rectify these issues.

## 3. Problem One: Imprudent Focus on the Medium

Researchers tend to agree that sexting occurs via digital communication technology. However, they seem to differ on the specific types of digital communication devices that they include under the 'sexting' umbrella. For example, some researchers only consider an activity 'sexting' when it occurs via mobile phone (e.g., Gordon-Messer et al., 2013 [33]); others include all mobile internet-enabled devices (e.g., smart phone, tablet; Drouin & Miller, 2016 [34]); others still include all mobile and non-mobile internet-enabled devices (e.g., desktop/laptop computer; Davis et al., 2016 [35]). Over time, most internet-based platforms became accessible through both mobile and non-mobile devices. This technological evolution makes it difficult to limit a definition of sexting to any single device. By focusing only on people's use of "mobile devices", researchers have necessarily excluded the ways that people use non-mobile devices to engage in essentially identical behaviours.

That is, people can send sexual content to others between and across mobile and static devices. For example, Person A can be using a desktop computer when they take and send a sexual image of themselves to Person B, who receives a notice first on their smart watch, and then the image on their phone. Moreover, Person A might also send a sexually explicit text message to Person B from their smartphone later that day. Most sexting research captures Person B's behaviour, and the second part of Person A's, but most studies will miss the technology-mediated sexual interaction that occurred at the beginning of the example. As such, researchers have an incomplete picture of people's technology-mediated sexual experiences. Moreover, communication technologies will continue to evolve; new devices will enter the market and current devices will become outdated. It is unclear whether the specific device is even relevant in an era where all internet-connected devices are so closely intertwined. This lack of clarity harkens to early questions about communication: is the medium itself an essential component of sexting, or is it the behaviour—sexual communication via technology—that is essential (e.g., McLuhan & Fiore, 1967 [36])?

Indeed, some researchers and theorists have argued that the behaviour is more important than the medium, especially when the goal is to create research that stands the test of time and shifts in technologies. Towards this end, Flanagin (2020) [37] argued that researchers need to focus on behaviours that people enact via technology, as opposed to specific technological features themselves. That is, the medium is important in-so-far as the behaviours it allows users to enact. Furthermore, researchers need to target the capabilities of technologies that traverse specific devices [37]. For example, people can send a sexual text message from many different devices; therefore, this is a behaviour that traverses technological tools. Communications researchers have long discussed the affordances of the Internet and mobile communication technologies on a variety of behaviours [38]. The need to focus on behaviours and affordances over specific media are well exemplified in the case of studying specific social media platforms. Many researchers have studied the different experiences that people have when using specific social media platforms, such as Facebook or Instagram. However, these types of platforms and how people use them have evolved rapidly over time. A study about people's experiences using Facebook in 2015 is not necessarily relevant today because the platform, and what users can do on it, is not the same. However, a study examining the impacts of having others 'like' your content could still be relevant, because this function is still common across many different social media platforms. Similarly, sexting is a behaviour that will evolve depending on what the technology affords; however, the essential behaviours that people conduct remain the same, regardless of the device they use to enact it.

The evolution of behaviours in researchers' definitions of sexting has also created overlap between sexting and other previously distinct behaviours such as phone sex and cybersex. A comparison of sexting and cybersex presents a good 'case study' of this overlap. In their 2017 systematic review of sexting, cybersex, and phone sex, Courtice and Shaughnessy [4] found that most researcher definitions of sexting and cybersex described essentially the same set of behaviours: sending, receiving, or exchanging sexual content via digital communication technology. For example, Drouin et al.'s (2013) [19] definition of sexting (transmission of sexual material via phone or Internet) and Delmonico's (1997) [12] definition of cybersex (exchanging sexual content over the Internet) are only distinguishable by the addition of the phone in Drouin's (a technology that was predominantly tied to a landline in 1997). It seems that researchers have simply adopted the term that is trending, rather than consider the behaviours that comprise the trend itself. In Figure 1, we present Google Trends data that represents the overall volume of people's searches for the terms "cybersex" and "sexting" over the past 17 years (2004–2021). This data suggests that, as the number of people's Google searches for 'sexting' increased worldwide (in 2009), the number of searches for 'cybersex' concurrently declined. The practice of adopting popular terms has led to redundancies and gaps in empirical and theoretical knowledge. Indeed, researchers who examine 'sexting' have rarely used existing knowledge and theory from researchers who examine 'cybersex'—and vice versa. This fragmented science

significantly hinders the ability to build a comprehensive and useful understanding of the core behaviour at the heart of both sexting and cybersex: exchanging self-created sexual content via internet-enabled technology with at least one other person.

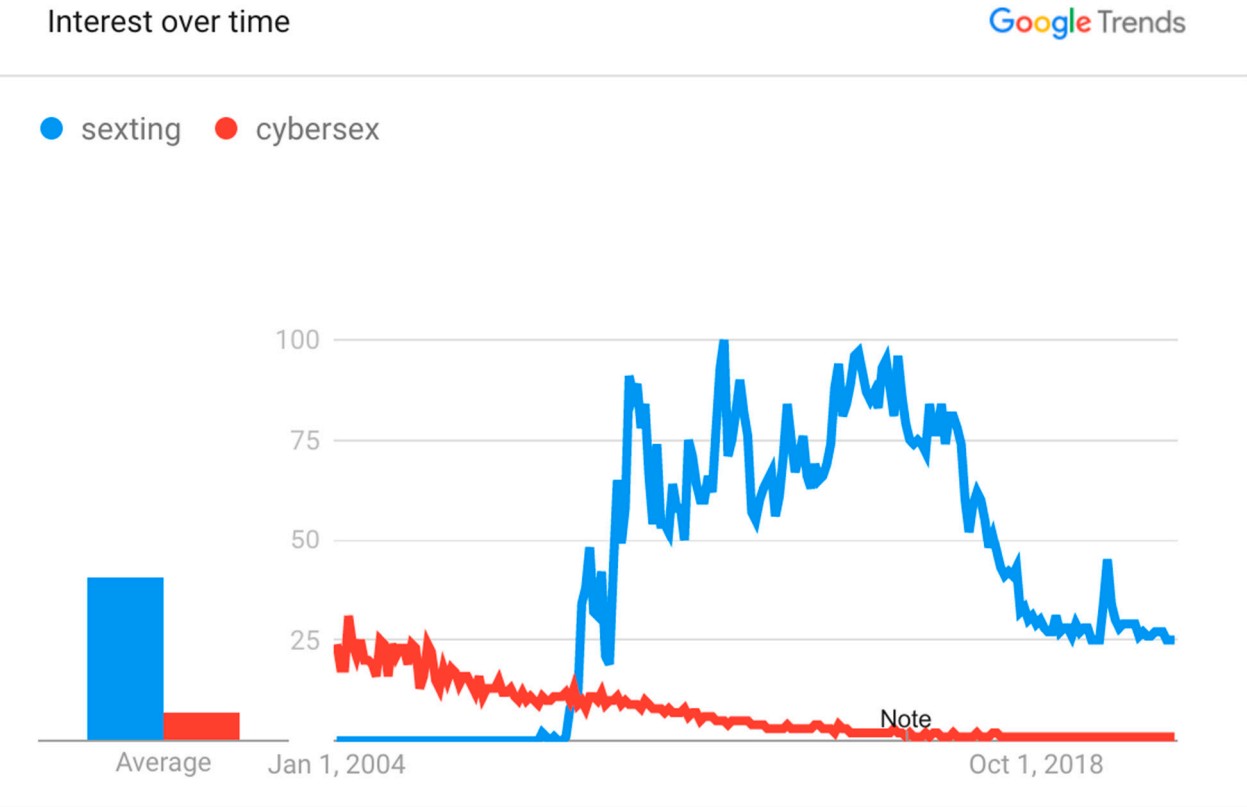

**Figure 1.** Google Trends output for Web search queries for the terms "sexting" and "cybersex" worldwide, from January 2004 to September 2021. Data source: https://trends.google.com/trends/explore?date=all&q=sexting,cybersex (accessed on 1 September 2021).

As communication technologies continue to evolve, it is likely that a new term will eventually supersede 'sexting' in popular discourse. It seems that 'sexting' gained linguistic notoriety because the term 'cybersex' did not automatically allow researchers to study the use of specifically mobile communication technology for sexual interactions. However, 'sexting' will eventually face the same issue; this term is unlikely to adapt to new, yet-to-be-conceived technologies and their corresponding terms. For example, virtual reality is a rapidly advancing technology that will allow people to have interactions with others—including sexual interactions—in virtual space. Like sexting and cybersex, a virtual sex interaction would involve an interpersonal sexual interaction that occurs via technology, perhaps in real-time. Because of the distinct technologies that virtual sex will require, it is unlikely that it will be referred to as sexting; instead, it is likely that laypeople will invent a new term to describe this type of sexual interaction. When a new term becomes popular, researchers who follow the trendy term may choose to 'reinvent the wheel' once again, instead of building on the knowledge we already have about sexting, cybersex, and other technology-mediated sexual interactions [21]. As an alternative, researchers could now begin to focus on people's behaviours rather than the technology they use to enact them. Unifying those behaviours under a single term will allow for a cohesive understanding of people's use of technology in their sexual lives, in the present and across time.

## 4. Problem Two: Inconsistent Conceptual Definitions

Conceptual definitions articulate the breadth and depths of behaviours that comprise an overarching construct or behavioural domain. A conceptual definition that clearly and accurately describes the content of a behavioural domain is the first step towards creating research measures with strong construct validity. Construct validity refers to the extent to which a given measure of a behavioural domain accurately quantifies the behavioural domain in question [39,40]. In other words, a conceptual definition refers to the behavioural domain that a researcher is interested in, whereas construct validity refers to how well that researcher's measure accurately tests people's experiences with that behaviour.

Construct validity is essential to building accurate and sustainable knowledge about a behaviour [39]. It occurs over a three-stage process. In the first "substantive phase", researchers define the construct by synthesizing researchers' and people's understandings of what the construct is—that is, how they define the construct in conceptual words. When researchers do not employ a strong conceptual definition of a construct, any attempt at measuring that construct will lack construct validity.

Unfortunately, there is little evidence that sexting researchers have created or adhered to a universal conceptual definition of sexting that accurately captures it from a behavioural domain or construct validation standpoint. Furthermore, there is little evidence that many of the proposed conceptual definitions of sexting apply across cultures outside of the Western context. Many researchers have noted inconsistencies in how sexting has been defined across studies with both adults and adolescents [2–8]. Specifically, the conceptual definitions used for sexting differ based on the inclusion/exclusion of particular (i) formats (e.g., text, images, videos), (ii) interaction dynamics (e.g., sending, receiving, forwarding, etc.), and (iii) sexual characteristics (e.g., full or partial nudity depiction of specific body parts).

Current researcher definitions of sexting include a wide variety of formats, such as text messages, audio recordings, photos, and/or videos. Across studies, researchers have varied in the types of media formats that they include in definitions of sexting. Some researchers have limited their definitions of sexting to include only text messages (e.g., Currin et al., 2016 [41]), whereas others have included only images (e.g., Crimmins & Seigfried-Spellar, 2015 [42]). Others have still included different combinations of activities in their sexting definitions. For instance, some researchers have described sexting as involving both photos and asynchronous videos (e.g., Gordon-Messer et al., 2013 [33]), and others have included text, photos, and asynchronous or live videos in their definitions (e.g., Dir et al., 2013 [43]; Drouin et al., 2013 [19]). Notably, most researchers have not included the exchange of audio messages (e.g., live conversation, voice memos), avatar sex (e.g., manipulating avatars on a screen or in virtual space), nor interaction via physical sensation (e.g., using teledildonic devices) [4]. Together, these varying conceptual definitions suggest that, for some researchers, sexting is a specific set of sexual activities, whereas, for others, sexting functions as an umbrella term that captures multiple—but not all—types of technology-mediated sexual activities.

There are various ways that people engage in sexual communication via technology: people can send content, ask someone for content, receive content, exchange content back and forth, share or forward content received from one person to another party, receive shared/forwarded content, watch other people exchange content, and more. Researchers have addressed the multitude of interaction dynamics that are available in sexting in a variety of ways. Some researchers have used conceptual definitions of sexting that focus on only one of the above actions, such as sending (e.g., Mitchell et al., 2012 [44]) or a back-and-forth exchange (e.g., Reyns et al., 2013 [45]). Others have used conceptual definitions of sexting that include several of the above actions; for example, Hudson et al. (2014) [46] defined sexting as "electronically sending, posting, or sharing/forwarding sexually explicit messages or seminude/nude images" (p. 183). There also seems to be an implicit debate in the literature about whether forwarding, sharing, or online posting sexts created by someone other than the forwarder/sharer/poster (presumably without

the original creator's consent) constitutes *sexting*, or is something altogether different [47]. Indeed, some researchers have included non-consensual forwarding, sharing, and/or posting behaviours in their definitions of sexting (e.g., Mori et al., 2020 [48]), and others have detailed that sexting involves sending a photo or video of yourself—implying that sexting excludes sharing content created by someone else (e.g., Ricketts et al., 2015 [49]).

Finally, researchers seem to disagree about which sexual characteristics must be present in order for a behaviour to constitute 'sexting'. Some researchers have created definitions of sexting that include sexual terminology—such as sexy, sexually explicit, or sexually suggestive [2]. Other researchers have defined sexting based on the amount of nudity present in images. In their systematic review and meta-analysis of sexting research, Madigan et al. (2018) [7] found that some researchers included only nude images in their definition of sexting, whereas others included depictions of both partial and full nudity. Of those who specify that sexting requires partial or full nudity, some researchers further specify which body parts must be visible to constitute sexting (e.g., breasts, genitals, buttocks) [2]. Of course, establishing the types or extent of nudity also then limits the definition of sexting to something that involves only images. Much like offline or in-person sexual activity (e.g., Peterson & Muehlenhard, 2007 [50]), there are a wide variety of characteristics that people may consider 'sexting'. However, researchers have not clearly delineated the line that technology-mediated behaviours must cross in order to definitively be 'sexting'.

The inconsistent conceptual definition of sexting across studies is inherently problematic. We are not the first to suggest that the large differences in sexting prevalence are likely due to definitions that create more or less opportunity for participants to endorse sexting [2–8]. Inconsistent definitions of sexting make it challenging—if not impossible—for researchers to synthesize understandings of sexting correlates, predictors, and outcomes across studies. For example, some researchers have found that sexting is related to negative emotional outcomes (e.g., Dake et al., 2012 [51]; Mitchell et al., 2012 [44]); others have found no association between sexting and negative emotional outcomes (e.g., Gordon-Messer et al., 2012 [33]; Hudson, 2011 [46]); others have found an association between sexting and positive emotional outcomes (e.g., Englander, 2012 [52]). Moreover, the inclusion and exclusion of different types of media and sexual characteristics are likely to have different potential impacts and outcomes. For example, sending sexual text messages may lead to different outcomes than would sending sexual images or videos. Similarly, having had a 'sext' non-consensually shared with a stranger no doubt has different outcomes relative to consensually sending a 'sext' to a romantic partner. Furthermore, when researchers consistently do not adhere to a strong conceptual definition, the entire area of study is called into question. Without a shared understanding of what sexting is (and is not), researchers cannot accurately measure, and therefore understand, people's experiences with, predictors of, and outcomes of the behaviour.

## 5. Problem Three: Poor Measurement Practices

It is only very recently that sexting researchers have begun to publish information on the development, reliability, and validity of measures of sexting. Most measures of sexting involve what Flake and Fried (2020) [53] describe as "questionable measurement practices" (QMPs). QMPs are the decisions that researchers make (or do not make) that raise doubts about the validity of a measure used in a study, and ultimately the study's final conclusions. Across studies, sexting researchers have employed several QMPs: reliance on ad hoc, single-item measures (instead of multi-item measures), not assessing internal or external validity, not using measures consistently, lack of item clarity, and not specifying context in measure instructions.

Many researchers have relied on ad hoc, single-item measures to examine people's sexting experience. This is likely because they are easier to create and lower in participant burden relative to multi-item measures. Some researchers used global, single-item measures in which they include the term 'sexting' in their question. For example, researchers

might ask, "Have you ever engaged in sexting?" (e.g., Walrave et al., 2015 [54]). These types of questions rely on all participants sharing the same understanding of the term 'sexting', thus leaving the term open to the interpretation of each individual participant. Shaughnessy and Byers (2013) [55] describe how people can make two types of errors when responding to single-item measures of sexual behaviours: (i) errors of commission and (ii) errors of omission. Errors of *commission* occur when people endorse experience with a particular activity, but their experiences do not fit with the researcher's conceptualization of that behaviour. For example, a researcher may conceptualize 'sexting' as an activity that only involves the exchange of sexual photos and videos. However, a participant who thinks that sexting includes exchanging sexually explicit text messages might say that they have engaged in sexting, even if they have never exchanged sexually explicit photos or videos with another person. This error of commission—a participant saying they have engaged in sexting when, by the researcher's definition, they have not—will likely cause the researcher to overestimate the incidence of sexual photo and video exchanges in their sample. Errors of *omission* occur when people fail to endorse experience with a particular activity, despite having had an experience that does fit with a researcher's conceptualization of that behaviour. Using the same example, a person who believes that only exchanging sexually explicit photos counts as sexting may not say that they have 'sexted', even if they have engaged in a sexual video exchange (which would in fact fit with the researcher's conceptual definition of sexting). This error of omission—a participant saying they have not engaged in sexting when, by the researcher's definition, they have—will likely cause the researcher to underestimate the incidence of sexual photo and video exchanges in their sample.

Researchers have also used single-item measures that examine only one 'type' of sexting. Kosenko et al. (2017) [6] examined the ways that researchers have measured sexting across studies, and identified several studies where researchers used single-item measures that addressed only one facet of sexting behaviour. For example, Temple and Choi (2014) [56] only asked participants if they had ever sent naked pictures of themselves to another person, whereas Jonsson et al. (2015) [57] asked participants if they had ever posted sexual pictures of themselves to the internet. Other researchers used single items that did not align with the conceptual definitions that they themselves delineated for sexting. For example, Gordon-Messer et al. (2013) [33] used the following conceptual definition for sexting: "sharing sexually suggestive photos or messages through cell phones and other mobile media" (p. 301). However, their measure only asked participants if they had ever sent and received a "sexually suggestive nude or nearly nude photo or video" of themselves or of someone they know (p. 302). These examples highlight the limited, if not poor, validity that single-item measures provide. Validity essentially refers to the accuracy with which a measure assesses what it intends to assess. Measures focused on a single facet of sexting, that do not align with the researcher's conceptual definitions of sexting are fundamentally problematic and raise questions about the validity of results and conclusions reliant on these measures.

Typically, multi-item measures provide a more accurate and psychometrically sound approach to measurement (e.g., Epstein, 1980 [58]; Nunnally & Bernstein, 1994 [59]). Multi-item measures for a behavioural group include a list of specific sexual activities that all fit within that domain. For instance, a researcher who uses a multi-item measure that includes separate questions about exchanging sexual (i) photos, (ii) videos, and (iii) text messages is less likely to encounter participant errors of commission and omission, compared to researchers who simply ask about 'sexting'. Moreover, multi-item measures allow researchers to calculate indicators of the measure's reliability and internal consistency. They also provide greater ability to evaluate the measure's validity across multiple dimensions, including content validity (that the measure captures the breadth of content relevant to the construct). Multi-item measures are less open to differences of interpretation across participants, because each question asks about one specific behaviour that fits within a behavioural domain. Additionally, aggregating multiple items decreases the impact of

bias and error. Therefore, any potential biases or errors will have less impact on both the reliability and validity of the measure—and ultimately the results and interpretations drawn from the findings.

Regardless of whether researchers have used single- or multi-item measures of people's sexting experiences, most have not directly assessed the structural and external validity of those measures. Structural validity refers to the extent to which the scores of a scale are an adequate indication of the construct being measured (i.e., using item analysis, factor analysis, reliability coefficients, and measurement invariance) [39]. External validity refers to the extent to which a given measure assesses the construct with other samples and in other contexts (i.e., assessing convergent, discriminant, and predictive variables, and examining differences between groups or samples) [39]. Both structural and external validity must be examined to ensure that a measure has construct validity; in this case, that a given scale is able to provide an accurate assessment of people's sexting experience. A key barrier to structural and external validity is the fact that researchers have not consistently used the same measures of sexting across studies. Instead, many researchers have opted to create their own measures that have rarely undergone any psychometric evaluations of their structural or external validity (for exceptions, see Molla Esparza et al., 2020 [60] and del Rey et al., 2021 [61]). This practice has resulted in many distinct single and multi-item measures of 'sexting'. Even when researchers use multi-item measures, the types of devices, format, interaction dynamics, and sexual characteristics vary from study to study [6]. Furthermore, the response anchors that researchers have used in their questions have differed greatly across studies. Some researchers have asked about lifetime sexting prevalence, while others have focused on the frequency of sexting in a limited time period. Because of these variations in measurement practices across studies, it is not possible to evaluate the construct validity of any one of these measures over time. It is also not possible to have a strong foundation of knowledge about sexting.

Finally, important contextual elements are blatantly absent from the majority of research on sexting. In particular, most researchers have not addressed the (i) relationship and (ii) consent context of people's sexting experiences. Some research has suggested that patterns of participation in sexting may differ depending on the type of relationship between the partners (e.g., whether they are in a committed romantic relationship versus whether they are strangers) [19]. Moreover, in studies that do include the relationship context of people's sexting experiences, researchers consistently find that the majority of people engage in sexting only with a committed romantic partner [1,19,62–65]. Furthermore, most researchers do not address the consent context of peoples' sexting experiences. Some research has suggested that, much like offline sexual activities, sexting can be (i) desired and consensual; (ii) undesired and consensual; and/or (iii) undesired and non-consensual [27,47,66,67]. It is also likely that people have different motivations for, and experience different outcomes from, engaging in consensual, compliant, and non-consensual sexting. Yet, unless the relationship or consent context is a key component of the study, researchers do not seem to report whether they have asked about sexting with any and all types of partners, or in any and all consent contexts. Therefore, it seems likely that researchers are not providing participants with clear instructions about the types of relationship or consent contexts to consider when responding to questions about sexting. Without clear instructions, some participants may (and others may not) consider particular relationship and consent contexts when responding to questions about sexting. This not only creates inconsistent response patterns, but also leaves the data open to researcher biases in interpreting the contexts in which sexting occurs—and further, whether or not it is 'problematic'. The only way to understand differences in sexting experiences and contexts is to provide clear instructions to ensure that participants understand the researcher's questions.

## 6. Problem Four: Lacking Theoretical Frameworks

Most of the research on sexting has very little connection to existing theoretical frameworks. In general terms, theories about human behaviour have two key components: they (i) describe a behaviour and (ii) make predictions about future behaviours. Theories are formulated and used in research to develop hypotheses and research questions; guide data collection; interpret the data; and propose explanations for observed phenomena across a range of contexts, timeframes, and people. When research is guided by theory, researchers are able to build on past knowledge and create immediate and sustained contributions to knowledge. Furthermore, atheoretical research is more susceptible to Type I errors—that is, publication of findings that will not replicate [68]. A theoretical framework lends structure to empirical research design, helps the researcher explain and provide context for their hypotheses, and can help to reduce potential researcher biases (for instance, focusing on the ways that sexting might be 'problematic' and disregarding potential positive outcomes). Yet, instead of conducting theoretically informed studies, researchers have focused on exploratory, atheoretical studies seeking to establish the prevalence and correlates of sexting among different demographic groups. For instance, both Kosenko et al. (2017) [6] and Van Ouystel et al. (2018) [8] identified only two studies that used a theoretical framework. This research is essential for understanding how widespread 'sexting' is and for whom it is more or less common. Moreover, this initial exploratory approach is in line with research on new technologies generally and is tied to a cycle of limited progress (see Orben, 2020 [21]). To better understand the role of technology in people's sexual lives and relationships, we must also explain why people engage in sexting, and what outcomes they experience. To do this, theoretical grounding is necessary. Ultimately, the lack of theoretical framing in sexting research makes it difficult to organize findings across studies in a coherent way, to explain particular phenomena, and to make connections between people's technology-mediated and in-person sexual experiences.

Using offline theories to guide sexting research has the benefit of connecting it to a broader understanding of human sexual behaviour and relationship patterns. Flanagin (2020) [37] argued that examining in-person experiences should be a starting point for researchers interested in technology-mediated human behaviours. Furthermore, theories about offline behaviours can spur further research into technology-mediated behaviours. Thus, researchers can and should apply well-developed theories of offline behaviour to sexting research. When researchers have used a theoretical framework to guide studies on sexting, they have relied on theories developed to explain offline sexual interactions and romantic relationships. For instance, researchers have used attachment theory [1,69], ambivalent sexism theory [67], social learning theory [70,71], theory of planned behaviour [54], and script theory [72] as frameworks to understand people's sexting behaviours. However, it is currently unclear if—and generally unlikely that—these theories perfectly apply in new, technology-based contexts. Therefore, researchers must also examine how these theories may be further developed to account for the different context(s) that technology affords. Indeed, sexting is not sexting without technology, yet researchers have not examined how the use of technology itself might play a role in people's experiences of sexting.

Sexting researchers have not explored nor applied theories that have been specifically developed for online or technology-mediated contexts. Indeed, this is a problem common to research on sexual behaviours in general. Sexuality researchers have been slow to integrate technology-relevant theories into their understanding of sexual behaviours and phenomena, despite the rapid integration of "sex tech" in people's everyday lives [73]. However, there are numerous theories on computer-mediated communication, information technology, human-computer interaction, and cyberpsychology that would also be of benefit to sexting researchers (see Shaughnessy & Braham, 2021 [73] for a brief overview). Using existing theoretical frameworks that account for technological context to inform sexting research would improve the quality of these studies and deepen our understanding of people's technology-mediated sexual behaviours. Application of theoretical frameworks with a technological lens would necessarily require researchers to integrate technology

considerations into all stages of the research process. For example, theories that integrate human behaviour and technology help to explain which technologies people use, how they use them, why they continue/discontinue use, and how technology-mediated behaviour relates to their offline experiences. Indeed, there are features of technology (e.g., anonymity, emotional expression, privacy, monitoring, etc.) that afford users something they cannot as easily access when engaging in offline sexual interactions. As such, using these theories will help sexting researchers (who typically come from disciplines that do not traditionally account for technology in human behaviour research) to identify which novel questions to examine and tech-specific variables to include. Furthermore, theories that consider and integrate technology may offer novel interpretations of research findings.

### 7. How Do We Move Forward?—Proposed Solutions

These four problems are not without solutions. We are not the first to propose solutions to the methodological problems of sexting research. For instance, researchers previously have called for better conceptual definitions of sexting, stronger measurement practices, and the use of existing theoretical models to inform studies and interpret results [2–8]. Furthermore, it appears that these recommendations have led to some positive changes in sexting research. For instance, more sexting researchers have used theory to guide their research [1,54,67,69–72]. However, there remain serious limitations for researchers who seek to create knowledge that will outlast the inevitable obsolescence of 'sexting'.

In light of the methodological limitations still present in sexting research, we propose that researchers stop studying sexting altogether. Instead, we propose that researchers shift to examining the behavioural domain of technology-mediated sexual interaction [4]. As we describe below, adopting TMSI will address all four problems in sexting research. Moreover, it will put researchers on the path of understanding behaviours rather than amorphous buzzwords. We also provide concrete direction for TMSI researchers who seek to improve their measurement practices, and to use technology-focused theoretical frameworks.

A clear and accurate conceptual definition of sexting would improve many of the problems in sexting research. Unfortunately, we do not believe that problems of defining sexting are surmountable. Sexting did not originate as a researcher-derived concept; instead, researchers interpreted and adopted the term from (predominantly Western) popular culture and media discourse. Researchers can and should examine novel phenomena that stem from societal trends and observations. However, in doing so, researchers should also consider layperson opinions on what sexting is and use that information to construct a conceptual definition that can then be measured in a reliable and valid way. However, this is not the approach that researchers have taken in understanding and defining sexting. Instead, researchers adopted many different and sometimes contradictory definitions of sexting, to the extent that it is now a nebulous term. Recently, a few researchers have created measures of sexting with a focus on construct validity [60,61]. However, even in this context, it is likely researchers and laypeople will always carry individual biases about what sexting is because of the origins of the term in relation to mobile phones.

To create knowledge that stands the test of time and evolving technology, researchers must target the basic capabilities of technology that traverse technological devices (see Flanagin, 2020 [37]). Almost all conceptual definitions of sexting do include at least one person sending (via technology) something sexual to at least one specific other person, who may or may not send one back. Thus, sexting includes sending, receiving, or exchanging sexual content with a specified other person, via technology. Sexting is also self-created and sexually explicit content. Indeed, that sexting involves sexual content that is self-created may be the key factor that distinguishes this behaviour from other types of online sexual activities, such as sharing pornography (featuring separate actors), or non-consensual for-warding of another person's sexts. Thus, researchers could simply create and consistently use a conceptual definition of sexting that includes all of these elements. However, this would still leave 'sexting' research vulnerable to becoming outdated when the devices that people use to communicate inevitably evolve, or when popular terminology changes in

particular cultural contexts. Furthermore, adopting this definition would keep sexting research detached from knowledge about other, related behaviours. The core components of sexting—engaging with at least one other person and self-created sexual content—clearly overlap with the core components of related behaviours. For instance, cybersex is the exchange (usually in real-time) of self-created sexual content and conversation between at least two people, via internet-connected devices. Phone sex (i.e., self-directed sexual conversation between at least two people over live audio), haptic sex (i.e., self-directed tactile sensations that can be sent and received between at least two people by control of a haptic device), and avatar sex (i.e., self-directed sexual activities and conversations that occur between at least two people, via personalized digital characters) also share the core components of self-created sexual content shared with another person. Thus, the clear similarities across terms highlight conceptual reasons for connecting these types of interpersonal, technology-mediated sexual behaviours under one area of inquiry.

**TMSI.** To improve upon and unify our understanding of the essential behaviour at the core of all these specific terms, we propose that researchers adopt the behaviourally-focused conceptual definition of technology-mediated sexual interaction. Technology-Mediated Sexual Interaction (TMSI) refers to any interpersonal interaction with a specified other person(s) that includes self-created, sexually explicit content and occurs through the use of communication technology [4]. As a construct, TMSI is focused on an overarching behavioural domain that includes 'sexting'. TMSI captures all types of interpersonal sexual interactions that occur via technology, as well as the different formats, interaction dynamics, and sexual characteristics that such interactions can include. As such, the definition of TMSI does not need to change when novel, not-yet conceived technologies emerge and are adopted for sexual interactions. Because of its focus on behaviour, TMSI is also likely to apply in all cultural contexts where technology is used for sexual interactions. By adopting TMSI, researchers will immediately be better able to examine the full range of behaviours that include and are similar to sexting. Furthermore, research on a behavioural domain will improve research methodology, knowledge, and knowledge synthesis on the many ways that people use technology for sexual interactions; how these are similar and different; what factors predict, explain, or modify varying TMSI experiences; and how all of these lead to diverse outcomes. TMSI research will unify sexting, cybersex, phone sex, avatar sex, haptic sex, virtual reality sex, etc., in the present and in the future.

The successful use of computer-mediated communication provides a strong example of a technology-focused behavioural domain. Computer-mediated communication (CMC) refers to any human communication that occurs through the use of two or more devices [74]. This overarching behavioural domain captures many popular terms for specific types of behaviours, such as emails, instant messaging, text messaging, video chat—and their more popular counterparts, such as 'texting', 'DMing', or 'FaceTiming'. By conceptualizing past, present, and emerging CMC under one umbrella, researchers were able to build on findings and theories even as technologies changed. Indeed, technological affordance theories—key components of much CMC research—are one of few technology focused theories that provide hypotheses, insights, and interpretation across specific types of technologies [21]. Thus, using a behavioural domain to connect distinct research topics is beneficial for researchers because it provides a common, organizing term that allows researchers to examine what is essentially the same underlying behaviour—even as the terminology and technology evolves. Indeed, CMC research is relatively long-standing—dating back at least to 1984 [75]. The successful use of CMC suggests that this same approach will work in the case of TMSI, which captures old and new terms (e.g., phone sex, cybersex, sexting, haptic sex) while maintaining conceptual integrity.

**Improve measurement practices.** The extent of our knowledge about people's sexting behaviours and experiences hinges on the validity of the tools that researchers use to measure sexting. Specifically, researchers need to shift to TMSI research by developing a measure of TMSI from a construct validation and measurement standpoint [39]. The following recommendations apply, regardless of the type of response anchors that

researchers use (e.g., frequency scales, yes/no checklist responses; for an example of a psychometric evaluation of a yes/no checklist measure, see DePasquale et al., 2019 [76]). The first "substantive" phase of measure development from this perspective includes a thorough review of previous literature to define the parameters of the construct and create an appropriate conceptual definition. In fact, Courtice and Shaughnessy (2017) [4] developed the conceptual definition of TMSI following a systematic literature review of sexting and cybersex research (note, there were no identified empirical studies on phone sex, avatar sex, virtual reality sex, or haptic sex at the time of their review). Researchers should then, in collaboration with other experts, create an item pool that taps into the different dimensions of TMSI. Ideally, these 'experts' would come from a variety of disciplines and perspectives, including sexuality and public health, as well as computer-mediated communication and cyberpsychology. These 'experts' should also include laypeople who have experience with sexting. Through qualitative research methods such as cognitive interviews [77] or other mixed-method approaches, researchers can gain valuable information from laypeople about how future participants might interpret items comprising a measure. In fact, researchers have completed this phase, presented results at peer-reviewed conferences [78], and are preparing the results for manuscript.

In the second, "structural" phase of measure development, researchers use quantitative analysis methods to examine the psychometric properties of the measure. These include item-level analysis, factor analysis, examination of reliability coefficients, and measurement invariance testing (see Flake et al., 2017 [39] for more details). In developing a measure of TMSI, this stage needs to occur in an initial sample of participants. In the third, "external" phase of measure development [39], researchers gather evidence for how well the measure does (or does not) relate to other constructs and predict criteria. This involves comparing the measure to other, existing measures to examine convergent and discriminant validity. Researchers also need to examine the structural aspects of any new measure in multiple samples, especially as characteristics of the sample differ from those of the measure development sample. Although there are few validated measures (and many non-validated measures) of sexting, most researchers have not examined these measures against each other. Furthermore, most researchers have not examined these measures in multiple, distinct samples that vary in demographic factors such as age, gender, culture, and race. Researchers who seek to create a measure of TMSI that has strong construct validity should plan to engage in ongoing and multiple studies, with diverse samples.

The procedures and best practices summarized by Flake et al. (2017) [39] are a result of decades of psychological research examining how best to measure behaviours [79–84]. Employing these 'best practices' is a time- and resource-intensive endeavor. However, this endeavor is also a necessary one; this is fundamental work that simply has not been rigorously or consistently applied in sexting research. To improve confidence in future knowledge that is generated about TMSI, researchers need to invest time into building TMSI measures with strong construct validity. In doing so, researchers need to avoid redundancies in research. Instead of creating multiple, slightly different TMSI measures, researchers should use the same measures of TMSI. Consistency in measurement across studies will lend itself to knowledge synthesization over time and across technologies and will prevent TMSI research from falling victim to the same measurement issues as sexting research has.

**Use theoretical frameworks in TMSI research.** The solution to the largely atheoretical lens that most researchers have applied to sexting research is, of course, to do the opposite. It is likely that the 'publish or perish' culture of academia has dissuaded many researchers from conducting high quality, theoretically-driven research. Indeed, high-quality research takes time and forethought to conduct; in comparison, it takes less time and effort to conduct exploratory research on ill-defined buzzwords, using 'on the fly' measures. When researchers take the time to consider and apply theoretical frameworks—especially frameworks that are technology-forward—we will see a considerable improvement in our empirical understanding of TMSIs (including sexting). Indeed, there are many fruit-

ful lines of inquiry left unaddressed as a result of inadequate application of theory in sexting research. As a starting point, Shaughnessy and Braham (2021) [73] recommend that TMSI researchers can make use of existing theory that accounts for technology, such as the Technology Acceptance Model (TAM) [85], the Unified Theory of Acceptance and Use of Technology (UTAUT) [86], and the Technology Integration Model (TIM) [87]. Researchers can also add new technology-focused elements to extend existing offline theories to the technology context. Finally, researchers may need to create new theoretical models that explain aspects of people's TMSIs that are unaccounted for in offline or non-sexual technology theories.

## 8. Conclusions and Future Implications

Despite over 10 years of research, we still know very little about people's sexting behaviours. The results of multiple literature reviews and meta-analyses have indicated that sexting research is marked by contrasting perspectives and contradictory findings, stemming largely from conceptual and methodological problems. Indeed, early sexting researchers did not take the time to agree upon a universal conceptual definition of sexting nor to develop and consistently use measures with strong construct validity. Furthermore, the definition of sexting remains amorphous, and will continue to be difficult to define because of the ever-evolving nature of communication technology across time and cultures. Thus, researchers need to pay less attention to 'sexting' and focus on the core, underlying behaviours that occur in sexting and other similar activities: technology-mediated sexual interaction. In doing so, researchers will develop a lasting behavioural understanding of people's use of technology for sexual interaction. We propose that researchers shift from sexting, cybersex, and other separate research streams to build a unified area of research on technology-mediated sexual interactions. This shift will necessarily incorporate sexting and its empirical findings. Researchers also need to move beyond exploratory research and develop a deeper, theoretically-driven understanding of people's TMSI behaviours. This theory-driven research should include and integrate existing frameworks that have a technological lens. People do and will continue to use technology for sexual interactions. In making the proposed changes, researchers will contribute a robust body of knowledge that stands the test of technological time.

**Author Contributions:** Conceptualization, E.L.C. and K.S; writing—original draft preparation, E.L.C.; writing—review and editing, E.L.C. and K.S.; visualization, E.L.C. and K.S. All authors have read and agreed to the published version of the manuscript.

**Funding:** Erin Leigh Courtice's work is supported in part by funding from the Social Sciences and Humanities Research Council.

**Institutional Review Board Statement:** Not applicable.

**Informed Consent Statement:** Not applicable.

**Data Availability Statement:** Not applicable.

**Conflicts of Interest:** The authors declare no conflict of interest.

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
