# Peer review of "Four Problems in Sexting Research and Their Solutions"

_sexes, doi:10.3390/sexes2040033_

Round 1

Reviewer 1 Report

Four Problems in Sexting Research and Their Solutions

This is a very interesting commentary on sexting research, focusing on their problems and their solutions.

I believe this is a relevant contribution to the readers of Sexes.

I think that these minor changes would improve the overall quality of the commentary:

  1. Please adhere to the journal reference format.
  2. Figure 1. I suggest authors to use source other than google. Google scholar aggregates many references that are not indexed.
  3. I suggest authors to include a section regarding “ethical implications”.
  4. Regarding “Problem Three: Poor Measurement Practices” authors should address the issue of the construct. If sexting is measured as a check list /Yes or No, how often, etc? this should be different than analyzing it as psychometrically validated construct.
  5. The commentary would benefit from a cross-cultural analysis on sexting, its meaning according to different cultures.
  6. I would suggest authors to include a “future implications” section.

Best wishes.

Author Response

Thank you for the reviews on our manuscript: 4 Problems in Sexting Research and Their Solutions. We appreciate the opportunity to revise our manuscript and we have taken into consideration all of the points that you have raised. Below we outline our response to each of the points raised.

I think that these minor changes would improve the overall quality of the commentary:

  • Please adhere to the journal reference format.
    • Thank you for this observation; we have updated the references in-text and in the reference list.
  • Figure 1. I suggest authors to use source other than google. Google scholar aggregates many references that are not indexed.
    • Thank you for pointing this out. This analysis is actually a representation of people’s Google searches for the terms “cybersex” and “sexting” over time, using Google Trends data. We have clarified our description of this data on page 4, as follows: “In Figure 1, we present Google Trends data that represents the overall volume of people’s searches for the terms “cybersex” and “sexting” over the past 17 years (2004 - 2021). This data suggests that as the number of people’s Google searches for ‘sexting’ increased worldwide (in 2009), the number of searches for ‘cybersex’ concurrently declined.” 
  • I suggest authors to include a section regarding “ethical implications”.
    • Thank you for this suggestion. We have added the following paragraph on page 3: “Around the world, research ethics have been guided by reports and declarations such as the Belmont Report and the Declaration of Helsinki. At the core of these and similar guidelines for ethical research are three fundamental principles: respect for persons, beneficence, and justice. To fulfill these baseline ethical standards, researchers must be committed to ‘do no harm’ and weigh the pros and cons of research projects before conducting them. During moral panics related to new technologies, public concerns are magnified and academic impact is heightened (Wartella & Robb, 2008). Therefore, scientific research that is informed by and predicated upon a moral panic can and does have far-reaching ethical implications. Unfortunately, some sexting research has been used to lend credibility to the ‘risk-forward’ narrative of sexting, especially about young people sexting. The ethical implications of this are clear; sexting research has had a profound impact on legislation that predominantly serves to threaten people who have sexted with legal penalties (e.g., broad application of child pornography charges in inappropriate contexts; Albury & Crawford, 2012). Undoubtedly, the intentions of these laws and related policies are to protect young people from harm. However, they ​​– and the research supporting a one-sided view of sexting – also inadvertently create harm. For instance, Setty (2019) suggested that the indirect impacts of these researcher-informed policies included social shaming and victim blaming of young people who have sexted, as well as a denial of rights to bodily and sexual expression. In the context of broad social pressures such as those created by moral panics, researchers must take additional precautions to consider the potential far-reaching harms to stakeholders prior to undertaking research (including exploratory research). It is now the responsibility of sexting researchers to not conduct and present research from a more balanced perspective, but also promote a balanced perspective amongst policy makers, legislators, educators, and the general public.  
  • Regarding “Problem Three: Poor Measurement Practices” authors should address the issue of the construct. If sexting is measured as a check list /Yes or No, how often, etc? this should be different than analyzing it as psychometrically validated construct.
    • Thank you for this observation. To address this recommendation, we have added the following on page 13: “The following recommendations apply regardless of the type of response anchors that researchers use (e.g., frequency scales, yes/no checklist responses; for an example of a psychometric evaluation of a yes/no checklist measure, see DePasquale et al., 2019).”
    • We have also added the following on page 9 to further address this recommendation: “Furthermore, the response anchors that researchers have used in their questions have differed greatly across studies. Some researchers have asked about lifetime sexting prevalence, while others have focused on the frequency of sexting in a limited time period.” 
  • The commentary would benefit from a cross-cultural analysis on sexting, its meaning according to different cultures.
    • Thank you for this suggestion. We have added the following throughout the manuscript to address possible differences in sexting definitions across cultures:
      1. Page 6: “Furthermore, there is little evidence that many of the proposed conceptual definitions of sexting apply across cultures outside of the Western context.”
      2. Page 11: “Sexting did not originate as a researcher-derived concept; instead, researchers interpreted and adopted the term from (predominantly Western) popular culture and media discourse.”
      3. Page 12: However, this would still leave ‘sexting’ research vulnerable to becoming outdated when the devices that people use to communicate inevitably evolve, or when popular terminology changes in particular cultural contexts.
      4. Page 12: “Because of its focus on behaviour, TMSI is also likely to apply in all cultural contexts where technology is used for sexual interactions.”
      5. Page 13: “Furthermore, most researchers have not examined these measures in multiple, distinct samples that vary in demographic factors, such as age, gender, culture, and race.” 
      6. Page 14: “Furthermore, the definition of sexting remains amorphous – and will continue to be difficult to define, because of the ever-evolving nature of communication technology across time and cultures.”
  • I would suggest authors to include a “future implications” section.
    • Thank you for pointing this out. We have modified the conclusion sub-heading to clarify that this section also contains implications for researchers interested in studying sexting and TMSI in the future. This subheading now reads “Conclusions and Future Implications”; the section itself includes the following information relevant for future research (page 14): “We propose that researchers shift from sexting, cybersex, and other separate research streams to build a unified area of research on technology-mediated sexual interactions. This shift will necessarily incorporate sexting and its empirical findings. Researchers also need to move beyond exploratory research, and develop a deeper, theoretically-driven understanding of people’s TMSI behaviours. This theory-driven research should include and integrate existing frameworks that have a technological lens. People do and will continue to use technology for sexual interactions. In making the proposed changes, researchers will contribute a robust body of knowledge that stands the test of technological time.” 

Reviewer 2 Report

I found this paper really an interesting point of view on sexting research that could give some guidelines in the next studies on this emergent topic.

The commentary has a good organization and it guides the reader in their perceptions of the problems in Sexting research and also on the possible solutions

I have only some minor comments.

I didn’t find a clear suggestion to solve the problem of the methodology. I can propose also the possible adoption of also a qualitative study design or a mixed-method approach (qualitative-quantitative). For example, software for possible narratives in participants’ interviews, such as N-Vivo or Spad-T or possible photographic/pictures material such as Atlas-T. The questionnaires should be validated and should reach different types of population by age and gender.

The next studies can start also from explorative/pilot studies adopting more qualitative approach, defining post a possible quantitative approach.

Author Response

Thank you for the reviews on our manuscript: 4 Problems in Sexting Research and Their Solutions. We appreciate the opportunity to revise our manuscript and we have taken into consideration all of the points that you have raised. Below we outline our response to each of the points raised.

I didn’t find a clear suggestion to solve the problem of the methodology. 

  • Thank you for pointing this out. Our methodological solutions are centered on researchers adopting and focusing on TMSI as a behavioural construct, rather than sexting. We have added the following throughout the Proposed Solutions section of manuscript to clarify this:
    • Page 11: “In light of the methodological limitations still present in sexting research, we propose that researchers stop studying sexting altogether.”
    • Page 12: By adopting TMSI, researchers will immediately be better able to examine the full range of behaviours that include and are similar to sexting. Furthermore, research on a behavioural domain will improve research methodology, knowledge, and knowledge synthesis on the many ways that people use technology for sexual interactions; how these are similar and different; what factors predict, explain, or modify varying TMSI experiences; and how all of these lead to diverse outcomes. TMSI research will unify sexting, cybersex, phone sex, avatar sex, haptic sex, virtual reality sex (etc) in the present and in the future.” 

I can propose also the possible adoption of also a qualitative study design or a mixed-method approach (qualitative-quantitative). For example, software for possible narratives in participants’ interviews, such as N-Vivo or Spad-T or possible photographic/pictures material such as Atlas-T. 

  • Thank you for this suggestion. In line with this and your other comment below, we have added the following to page 13: “These ‘experts’ should also include laypeople who have experience with sexting. Through qualitative research methods such as cognitive interviews (Willis, 2004) or other mixed-methods approaches, researchers can gain valuable information from laypeople about how future participants might interpret items comprising a measure.”

The questionnaires should be validated and should reach different types of population by age and gender.

  • Thank you for this recommendation. We have updated our recommendations for better measurement practices on page 13, as follows: “Furthermore, most researchers have not examined these measures in multiple, distinct samples that vary in demographic factors, such as age, gender, culture, and race. Researchers who seek to create a measure of TMSI that has strong construct validity should plan to engage in ongoing and multiple studies, with diverse samples.” 

The next studies can start also from explorative/pilot studies adopting more qualitative approach, defining post a possible quantitative approach.

  • Thank you for this suggestion. We have clarified our recommendations related to using qualitative methods to inform better measurement practices on page 13, as follows: “These ‘experts’ should also include laypeople who have experience with sexting. Through qualitative research methods such as cognitive interviews (Willis, 2004) or other mixed-methods approaches, researchers can gain valuable information from laypeople about how future participants might interpret items comprising a measure.”